# Histoplasmosis in Solid Organ Transplantation

**DOI:** 10.3390/jof10020124

**Published:** 2024-02-02

**Authors:** Nicolas Barros, L. Joseph Wheat

**Affiliations:** 1Department of Medicine, Indiana University School of Medicine, Indianapolis, IN 46202, USA; 2Division of Infectious Diseases, Indiana University Health, Indianapolis, IN 46202, USA; 3Miravista Diagnostics, Indianapolis, IN 46241, USA; jwheat@miravistalabs.com

**Keywords:** histoplasmosis, *Histoplasma*, solid organ transplant, transplant

## Abstract

*Histoplasma capsulatum*, the etiological agent for histoplasmosis, is a dimorphic fungus that grows as a mold in the environment and as a yeast in human tissues. It has a broad global distribution with shifting epidemiology during recent decades. While in immunocompetent individuals infection is usually self-resolving, solid organ transplant recipients are at increased risk of symptomatic disease with dissemination to extrapulmonary tissue. Diagnosis of histoplasmosis relies on direct observation of the pathogen (histopathology, cytopathology, and culture) or detection of antigens, antibodies, or nucleic acids. All transplant recipients with histoplasmosis warrant therapy, though the agent of choice and duration of therapy depends on the severity of disease. In the present article, we describe the pathogenesis, epidemiology, clinical manifestations and management of histoplasmosis in solid organ transplant recipients.

## 1. Introduction

*Histoplasma capsulatum* is a dimorphic fungus that grows as a mold in the environment and as a yeast in human tissue. Historically, the genus *Histoplasma capsulatum* has been classified by its geographic distribution, morphology, and clinical symptoms. This classification includes three varieties: *var. capsulatum*, *var. duboisii* and *var. farciminosum* [1]. Human histoplasmosis occurs due to infections with *H. capsulatum var. capsulatum* or *H. capsulatum var. duboisii*. The latter only occurs in small pockets of Africa [2]. More recent studies have shown that the genus *Histoplasma* is a complex of cryptic species with multiple genetically distinct genetic groups. There are five discrete genetic clusters (Nam1, Nam2, LAmA, Panama and Africa) [3]. In the current article, we will describe histoplasmosis based on its historical classification, though we acknowledge that this is an incomplete and old classification. Given that most cases are caused by *H. capsulatum var. capsulatum*, in the current review article, we will refer to it as *H. capsulatum*.

*H. capsulatum* has a worldwide distribution with presence in every continent except Antarctica. In North America, the areas with highest endemicity include the valleys of the Mississippi and Ohio rivers, where the incidence is estimated to be 6.1 cases per 100,000 individuals and almost 90% of the population will be exposed to histoplasmosis during their lifetime [4,5]. In a study from a hyperendemic area, 24% of the solid organ transplant candidates had evidence of prior exposure to histoplasmosis [6]. During past decades, there have been significant changes in the epidemiology of histoplasmosis [7]. This may be related to disruption of the environment, climate change, anthropomorphic activities and increased immunosuppressive conditions, including solid organ transplantation [8].

Seroprevalence in Central and South America has a wide range of regional variability, but prior exposure may be as high as 30% of the population [9,10]. While in most immunocompetent patients histoplasmosis is usually a self-resolving infection [11,12], solid organ transplant recipients and other immunocompromised hosts have a propensity to developed severe forms of the disease with dissemination to extrapulmonary organs [13]. Here, we will describe important aspects of human histoplasmosis in solid organ transplantation.

## 2. Pathogenesis

*H. capsulatum* can be found in microfoci of soil enriched with nitrogen with pH levels of 5–10. This environment is present in areas were copious amounts of bird excreta or bat guano is present [14,15].

In the environment, the mycelial phase has two types of conidia. The macroconidia measures 8–14 µm in diameter and displays distinct tuberculate projections (ship’s wheel). The microconidia measure 2–5 µm in diameter and are considered an infectious form. Given their size, microconidia can easily aerosolize with any disruption of their microenvironment (e.g., farming, demolition of old buildings, exposure to chicken coops, caves, etc.) [16].

The onset of infection occurs following inhalation of microconidia or mycelial fragments which reach the pulmonary alveoli. A temperature shift leads to conversion into the pathogenic yeast form. Upregulation of the proteins from the RYP family (Ryp1, Ryp2, Ryp3 and Ryp4) regulate the fungal morphology in response to temperature [17].

Upon reaching the alveoli, *H. capsulatum* must evade soluble pattern recognition receptors that display direct fungicidal activity. The collectins, surfactant protein A and surfactant protein D, have been shown to have direct fungicidal activity through a calcium-dependent mechanism of yeast permeabilization [18]. Surfactant protein A has been shown to be substantially reduced in patients co-infected with HIV and *H. capsulatum,* which may be associated with a higher fungal burden [19]. While the role of surfactant protein A and macrophage activation in diverse infectious pathogens has been studied in immunocompetent hosts, little is known about SP-A and SP-D in solid organ transplant recipients. In lung transplantation, low levels of surfactant protein A in exhaled particles has been associated with an increased risk of bronchiolitis obliterans [20].

*H. capsulatum* yeast is internalized by macrophages, and to a lesser extent, by dendritic cells and neutrophils, using three different complement receptors (CR3, CR4 and LFA-1) [11]. The Heat shock Protein 60 (Hsp60) of *H. capsulatum* is a major surface adhesin to macrophages and binds to the β subunit (CD18) of CR3 (CD11b/CD18) [12]. Macrophages can also respond to *H. capsulatum* by recognizing other pathogen-associated molecular patterns through pattern recognition receptors, including Dectin-1 [21]. This C-type lectin recognizes 1-3-β-D-glucan, which is a major polysaccharide in the cell wall of most fungal pathogens. While the interaction of Dectin-1 and 1-3-β-D-glucan is not directly involved in phagocytosis, it mediates pro-inflammatory cytokine expression (TNF-α and IL-6) in a mechanism dependent on CR3 association [22].

In order to achieve successful internalization without triggering a proinflammatory cascade, *H. capsulatum* yeasts must avoid the interaction of Dectin-1 and 1-3-β-D-glucan without compromising Hsp-60 and CR3 engagement. To accomplish this, *H. capsulatum* expresses an outer layer of 1-3-α-glucan, concealing 1-3-β-D-glucan which, in turn, impairs the recognition by Dectin-1 [23]. Furthermore, *H. capsulatum* yeasts express 1-3-β-glucanase (Eng1 protein) which hydrolyzes exposed 1-3-β-D-glucan that has not been concealed by 1-3-α-glucan [24]. The H antigen is a beta-glucosidase that remodels the yeast cell wall and is involved in nutrient acquisition [25].

Once in the phagosome, *H. capsulatum* evades both reactive oxygen and nitrogen species by secreting a Cu/Zn-type superoxide dismutase 3 (Sod3), Catalases (CatB [M antigen]) and CatP) and nitric oxide reductase (Nor1) [26,27,28,29]. Phagolysosome acidification is subsequently inhibited by the enzyme HMG CoA lyase [30].

Macrophages play a dual role by containing infection and providing an environment where replication and/or dormancy can occur. *H. capsulatum* uses them as a trojan horse for dissemination. Their infection may induce granuloma formation in different tissues which are, in turn, required to contain fungal growth and dissemination while protecting from widespread collateral inflammatory tissue damage [31]. Macrophages represented 70% of all cells in a murine model of *Histoplasma*-induced granulomas. Dendritic cells and T cells were also present within the granulomas. Dendritic cells provide the link between the innate and adaptive immune responses by acting as antigen-presenting cells [32]. Human dendritic cells drive CD4+ T cell differentiation and proliferation. CD4+ T cell polarization into the Th1 phenotype leads to the production of IFN-γ and TNF-α which activates intracellular killing and is pivotal for controlling infection [33]. CD4+ T cells are the main producer of IFN-γ and TNF-α within the granulomas [31].

Granuloma formation may be impaired in the event of immunosuppression which can lead to dissemination. Blockage of TNF-α has been associated with increased fungal burden and disseminated disease [34]. While about 90% of people exposed to *H. capsulatum* remain asymptomatic or have mild self-resolved symptoms, some develop severe histoplasmosis [16]. Therefore, the outcome of infection and clinical presentations depends on the interplay and delicate balance between the host’s innate and adaptive immunity and the fungal virulence factors (Figure 1).

## 3. Epidemiology and Risk Factors

In North America, the geographic distribution was first described in the 1950s and 1960s using antigen skin testing of persons living in a single county with no history of travel [5,35]. Since then, the geographic distribution of histoplasmosis has expanded to regions beyond their historical border (Figure 2) [36]. This may be related to changes in climate and human land use, with increased cultivation in several regions and urbanization of others [8]. Using a weighted analysis of three environmental soil characteristics measured across the continental United States, Maiga et al. showed that the soils suitable for growth of histoplasmosis have expanded into the upper Missouri River Basin, with pockets of suitability in almost all states [15].

Transplant-associated histoplasmosis can occur as a de novo infection from environmental exposures, as a donor-derived infection or as reactivation from a latent infection, though the latter remains controversial [37]. In a prior autopsy series in which calcified granulomas were examined, two thirds had yeast forms consistent with *Histoplasma* spp. However, fungal cultures remained sterile and inoculation in experimental animal models failed to cause infection which refutes the possibility of reactivation from prior infections [38]. Histoplasmosis has been reported after several years after returning from an endemic country, though it is possible that the patient in question may have had a low-level infection that never cleared but progressed following an increased net state of immunosuppression [39].

Despite their increased susceptibility to fungal pathogens, infections due to dimorphic fungi remain uncommon in solid organ transplant recipients. Of the dimorphic fungi, *Histoplasma* is the most common pathogen leading to clinical disease following solid organ transplantation. In a large prospective surveillance study among transplant recipients at 23 centers throughout the United States between 2001 and 2006 (Transplant-Associated Infection Surveillance Network, TRANSNET), histoplasmosis represented 75% of all dimorphic fungal infections, with a 12 month cumulative incidence rate of 0.102%. About 50% of all the infections occurred in kidney transplant recipients, though this is likely a representation of the increased number of kidney transplant recipients in the underlying cohort. The median time from transplantation was 274 days, and 15% of the patients with histoplasmosis died within 3 months of diagnosis [40].

In a more recent cohort of 24 solid organ transplant (SOT) centers between 2003 and 2010, Assi et al. identified 152 cases of histoplasmosis which occurred at a median time from transplantation of 27 months, though over a third occurred during the first year [13].

Another group showed that the global incidence of post-transplantation active histoplasmosis was 1 case per 1000 person-years [41]. The incidence was highest among pancreas transplant recipients (10.7 cases per 1000 person-years) followed by kidney–pancreas (2.6 cases per 1000 person-years), lung (2 cases per 1000 person-years), heart (1.2 cases per 1000 person-years), kidneys (0.6 cases per 1000 person-years) and liver transplant recipients (0.4 cases per 1000 person-years).

Several reports indicate that the rate of infections may be increasing. Freifeld et al. showed a 17-fold rise in histoplasmosis cases among SOT patients during 2002–2004, compared with the prior 5 year period [42]. This may be related to increased awareness and improved diagnostics. Benedict et al. showed that from 2001 to 2012, histoplasmosis-associated hospitalizations in transplant recipients had an annual percentage increase of 12.9% [43].

The risk factors for post-transplant histoplasmosis include increased net state of immunosuppression and environmental exposures. Of note, exposure to birds, bats or their droppings are described in more than 75% of outbreaks, but only 25% of the sporadic cases [36]. Given that over 95% of the cases are sporadic, the vast majority of patients do not recall having a clear exposure [36]. In another study of enhanced surveillance in nine states during 2018 and 2019, the patients reported exposures including handling plants (48%) and bird or bat droppings (24%). However, 22% of the patients did not report any specific exposures [40].

Data on the epidemiology of histoplasmosis in SOT outside North America are limited. In a large retrospective study of 1754 liver or kidney transplant recipients in Brazil, Batista et al. reported that the rate of infection was 0.2% [44]. Another Brazilian cohort showed that histoplasmosis was the second most common invasive fungal infection following Cryptococcosis and occurred in 1.1% of all their kidney transplant recipients. The median time to diagnosis was 24 months (IQR 11 to 38 months). The use of anti-thymocyte immunoglobulin and methylprednisolone pulses was associated with earlier cases of infections [45]. In a study of 102 autopsies of kidney transplant recipients in Brazil from 1968 to 1991, histoplasmosis was the underlying cause of death in 3% [46].

## 4. Clinical Presentation

While most immunocompetent individuals will have asymptomatic infections or self-resolved mild symptomatic disease, solid organ transplant recipients usually present with significant symptomatic disease [47]. Clinical presentations may be highly variable, non-specific and with a wide range of severity (Table 1). This often leads to a significant delay in diagnosis. Diagnosis is usually delayed by 1–3 weeks following initiation of symptoms [41,42,48]. Most patients experience fever (89–93%), shortness of breath and cough (57%) but symptoms may also include fatigue, diaphoresis, headache and weight loss (all). Histoplasmosis has also been identified as a cause of fever of unknown origin [49].

In a large cohort of 125 cases of histoplasmosis in SOT, only 8% of the patients presented with mild disease, while 63% presented with moderate disease, and 27% had severe disease requiring intensive care management [13]. In a smaller cohort of 23 SOTs, the severity of disease at presentation was almost identical (4%, 61%, 35%, respectively) [50]. Disseminated disease is the most common presentation, occurring in 81% of patients with no difference in severity related to the type of transplanted organ [13]. The lungs are most often involved (81%), regardless of the type of presentation (pulmonary vs disseminated) [13]. Pulmonary infiltrates were identified in 50–70% of the patients by chest X ray but 87–100% by chest CT. The most common type of infiltrates include multiple nodular disease (with or without miliary pattern 6%) and diffuse bibasilar infiltrates. Splenomegaly and lymphadenopathy were present in 50% of cases [41,50].

Other organs involved included bone marrow (32%), liver (22%), spleen (13%), gastrointestinal (11%), central nervous system (7%) and skin (3%) [41]. Interestingly, liver allograft was associated with 80% of the liver transplant recipients with histoplasmosis [48,51].

Laboratory abnormalities are frequent. These include pancytopenia, transaminitis, elevated lactate dehydrogenase, hypercalcemia and elevated inflammatory markers (e.g., ferritin) [41,52,53,54]. Histoplasmosis-induced hemophagocytic lymphohistiocytosis (HLH) has been reported [55]. This syndrome is the result of defects in NK cell cytotoxicity resulting in a positive feedback loop of uncontrolled intracellular infection, ongoing immune activation, and a lack of immune downregulation. HLH is characterized by the presence of fever, hepatosplenomegaly, cytopenias, hyperferritinemia, elevated soluble IL-2R and decreased NK function. The optimal treatment is unknown, though most clinicians only treat underlying infection.

## 5. Diagnostics

Diagnosis of histoplasmosis relies on direct observation of the pathogen (histopathology, cytopathology, and culture) or detection of antigens, antibodies, or nucleic acids (Table 2).

### 5.1. Culture, Histopathology and Cytopathology

Isolation of *H. capsulatum* from clinical specimens remains the gold standard for diagnosis of histoplasmosis, though histopathologic or direct microscopic identification are also considered definitive diagnosis [56]. However, having a trained pathologist/microbiologist is imperative for accurate reports.

Fungemia is present in 63% of patients and is higher in patients with disseminated compared to mild-to-moderate disease (90% vs. 49%). Lung or respiratory cultures were positive in 72% of patients and were more common in disseminated compared to mild-to-moderate disease (77% vs. 60%) [13]. Direct visualization of organisms in bone marrow occurred in 71% of patients [13].

### 5.2. Antigen

Detection of *H. capsulatum* polysaccharides in urine and serum was first developed in 1986 as a sandwich radioimmunoassay. Since then, it has been improved to a second-generation EIA, which allowed for semiquantitative results and reduced the number of false-positive results caused by human anti-rabbit antibodies, and subsequently, a third-generation quantitative test with improved sensitivity (MVista *Histoplasma* galactomannan EIA, Indianapolis, IN, USA) [57].

In a study of 18 solid organ transplant recipients with proven histoplasmosis from two university medical centers in the Midwest, antigenuria was present in all the cases [42]. Another report from an endemic area, showed that 12/14 (85%) of SOT patients with disseminated disease had a positive urine antigen test [41]. The lower sensitivity in this study may be related to the use of an older generation test for antigen detection as the study spanned from 1997 to 2007.

In a large multicenter study involving 152 SOTs (kidney: 51%, liver: 16%, kidney/pancreas: 14%, heart: 9%, lung: 5%, pancreas: 2%, others: 2%) with histoplasmosis, Assi et al. showed that urinary antigen testing was the most sensitive test, being positive in 93% of all patients [13]. Antigenemia was present in 86%. However, the test performed differently depending on the burden of disease. Antigenuria was present in 73% of the patients with pulmonary disease vs. 97% of those with progressive disseminated disease (*p* = 0.01). In addition, antigenemia was present in 59% of the patients with pulmonary disease vs. 89% of those with progressive disseminated disease (*p* = 0.03) [13].

While the MVista assays are laboratory-developed tests requiring processing in a central laboratory, an in vitro urine *Histoplasma* galactomannan EIA by IMMY (IMMY Alpha EIA, Norman, OK, USA) was FDA approved in 2007 and has the advantage of being available for use at local facilities. This is particularly useful in settings outside the United States. The IMMY alpha EIA has a sensitivity of 67% for progressive disseminated histoplasmosis in patients living with HIV [58]. Subsequently, IMMY developed a new platform using an analyte-specific reagent *H. capsulatum* antigen EIA (Clarus IMMY Histoplasma GM EIA, IMMY, USA). This in vitro test has been FDA cleared [59]. In their initial reports, the Clarus IMMY had a sensitivity of 64.5%. with a cutoff of >0.5 ng/mL which increased to 80.7% if the cutoff was lowered to >0.15 ng/mL [60]. The specificity at those cutoff levels was 99.8% and 96.3%, respectively. In a reevaluation of those cutoff levels, Theel et al. showed an improved positive and negative agreement with MVD EIA (82.3%). Of note, all indeterminates result were removed from the analysis [61]. A subsequent study from clinical samples, showed it had a lower overall sensitivity compared to MVista EIA (72% vs. 96%) [62]. Of note, both tests detected all patients with progressive disseminated histoplasmosis. A more recent study in patients living with HIV/AIDS with progressive disseminated disease showed that the sensitivity of Clarus IMMY EIA was 91.3% with a specificity of 90.9% [59]. It is unclear if the differences observed in the prior studies are related to study populations (immunocompetent vs. immunocompromised) and disease burden (pulmonary histoplasmosis vs progressive disseminated histoplasmosis).

The detection of *Histoplasma* galactomannan in other body fluids including bronchoalveolar lavage and cerebrospinal fluid has been described for the diagnosis of pulmonary and central nervous system histoplasmosis, respectively. Of note, the galactomannan tested in body fluids of patients with histoplasmosis is identical to galactomannan detected in blastomycosis [63]. *Histoplasma* galactomannan was detected in 84% of cases of pulmonary histoplasmosis and 83.3% of patients with pulmonary blastomycosis. The test had a specificity of 98% in patients with other pulmonary infections [64]. The sensitivity of the test in CSF for central nervous system histoplasmosis was 78% with a specificity of 97% [65].

Recently, a *Histoplasma* galactomannan antigen qualitative lateral flow assay was developed (MVD *Histoplasma* LFA). The test can be read manually and by an automated reader. It has an overall sensitivity of 78.8% with a specificity of 99.3%. While positive in only 50% of patients with pulmonary histoplasmosis, it was positive in 91.3% with disseminated histoplasmosis [66]. Furthermore, its sensitivity in immunocompromised patients was 93.6%.

In a study of patients living with HIV and progressive disseminated histoplasmosis in Mexico, the sensitivity of the MV LFA was 90.4%, which was similar to Clarus IMMY EIA (91.3%) [59]

In a study by Caceres et al. in Colombian patients living with HIV, the *Histoplasma* LFA showed an excellent analytical performance, with a sensitivity of 96% and specificity of 90% [67,68].

OptimumIDX recently developed another lateral flow platform (OIDx *Histoplasma* LFA). The test had a sensitivity of 92% but a specificity of 32% [69]. In a study from Ghana, the OIDx *Histoplasma* LFA had a positive concordance with Clarus IMMY of 98.4%, though negative concordance was not reported [70].

The galactomannan detected in the MiraVista antigen assay has almost the same molecular characteristics as those in *Blastomyces dermatidis*, *Paracoccidioides braziliensis* and *Talaromyces marnefii* [71]. Hence, they express almost complete cross-reactivity. Low-level cross-reactivity occurs with the galactomannan detected in coccidioidomycosis and other dimorphic fungi [71,72,73,74,75,76,77].

### 5.3. 1-3-β-D-Glucan

1-3 β-D-glucan is a component of the cell wall of most fungi and its detection in serum has been used to identify patients with invasive fungal diseases [78]. Previous studies have shown that serum 1-3-β-D-glucan is present in over 85% of all patients with histoplasmosis [79,80]. However, the lack of specificity and high false-positive rate in patients without an invasive fungal infection limits its use as a diagnostic tool for histoplasmosis.

### 5.4. Aspergillus Galactomannan

The test for the detection of *Aspergillus* galactomannan (Platelia, Bio-Rad, Marnes-La-Coquette, France) is widely used to diagnose invasive aspergillosis [81]. Several studies have shown that 67–100% of patients with disseminated histoplasmosis have a positive *Aspergillus* galactomannan test [82,83,84]. In an experimental model of invasive pulmonary aspergillosis, specimens that were positive for *Aspergillus* galactomannan (including those with very high levels) were negative for *Histoplasma* antigen EIA [84]. Cross-reactivity has also been noted with paracoccidioidomycosis (50%) and cryptococcosis (50%) [82].

Of note, patients with aspergillosis do not have positive urine or serum *Histoplasma* antigen EIA [84]. For that reason, it is important to rule out histoplasmosis and other fungi in the setting of a positive *Aspergillus* galactomannan.

### 5.5. Serology

The most common tests assessing antibodies against histoplasmosis include complement fixation (CF), immunodiffusion (ID) and ELISA immunoassay (EIA). CF detects antibodies against two antigens: histoplasmin and a yeast antigen. Historically, titers of 1:32 or above are suggestive of active histoplasmosis, while titers of 1:8 and 1:16 are considered presumptive evidence of histoplasmosis. A four-fold increase is suggestive of progression of the disease [85].

CF fixation or ID were positive in 33% to 36% of all cases, but detection of antibodies by CF was the basis for diagnosis in only 1 of 142 cases in the Assi report [13,41]. While 62% of the pulmonary cases were positive, only 28% of the disseminated cases were positive [13].

The ID test detects the presence of antibodies against antigens M and H. The M band is detectable in most immunocompetent patients with histoplasmosis (80%) but persists over time and does not distinguish between prior exposures or active disease. The presence of antibodies against H antigen detects active infection, though it lacks sensitivity (20%) [86].

ID and CF are complex and less standardized than EIA and not widely available in clinical laboratories. Prior reports indicate that antibodies by ID and CF are positive in 36–70% of cases [13,73]. In those reports, the authors do not provide the individual sensitivity of each test. However, data in solid organ transplant recipients are lacking. In patients with solid organ transplantation, antibody detection by EIA was positive in 50% of the patients (unpublished data).

Combining the detection of antigens and antibodies by EIA increased the sensitivity for the diagnosis of pulmonary histoplasmosis from 67% for antigen alone and 89% for antibody alone to 96.3% [87]. However, this cohort did not include patients with solid organ transplantation. The combination of detecting antigen and antibodies by EIA has also been studied in cerebrospinal fluid of patients with central nervous system histoplasmosis. In that study, Bloch et al. showed that the sensitivity of the test increased from 78% for antigen alone and 82% for antibody alone to 98% [88].

Of not, antibodies are detectable only 4–8 weeks after the initial infection and have lower sensitivity in the immunocompromised hosts [86].

### 5.6. Molecular Tests

Currently, there are no FDA-approved assays to detect histoplasmosis from human samples. However, a chemiluminescent DNA probe (AccuProbe test; Gen-Probe Incorporated 2011) is commercially available which decreases the time to identify the pathogen growing in culture [89]. A proteome-based technique, matrix-assisted laser desorption ionization time of light mass spectrometry (MALDI-TOF MS), has been shown to be highly accurate for the identification of *H. capsulatum* from cultures. This test can identify yeast forms and early mycelial cultures [90]. Of note, MALDI-TOF is only useful for the identification of *Histoplasmosis* in instruments with Vitek MS v3.0 database but not on MALDI Biotyper, Bruker Daltonics. The latter was unable to identify dimorphic fungi due to the lack of the fungal reference spectrum in its database [91].

There are multiple laboratory-developed *Histoplasma*-specific PCR assays, though clinical performance has not been described. Broad-range PCR of fungal 28S ribosome and internal transcribed spacers (ITS) have been used for the isolation of histoplasmosis in different tissues (e.g., paraffin-embedded biopsies) [92]. In recent years, metagenomics next-generation sequencing has been used for the identification of rare infections [93]. Recently, this assay was able to correctly identify two different cases of disseminated histoplasmosis [94].

## 6. Management

While most immunocompetent patients do not require therapy, all solid organ transplant recipients with histoplasmosis require therapy [47]. The agent of choice and duration of therapy depends on the severity of the clinical presentation (Table 3).

Most of the data have been extrapolated from trials in patients living with HIV. In patients presenting with disseminated disease, amphotericin B is associated with an increased clearance of cultures and a more rapid decline in median antigen levels in serum [95]. Liposomal amphotericin B is associated with less toxicity and mortality and is considered the agent of choice for induction therapy in patients with severe disease [96]. The American Society of Transplantation Infectious Diseases Community of Practice recommends continuation of induction therapy for a minimum of 1 to 2 weeks with clinical evidence of improvement [37]. In patients with CNS involvement, longer courses of induction therapy are recommended (i.e., 4 to 6 weeks) [37,97]. Following completion of induction therapy, therapy should step down to oral itraconazole. While the concomitant use of itraconazole and amphotericin B has not been associated with any antagonistic effects, in animal models, fluconazole has been shown to antagonize amphotericin’s reduction of fungal burden [98]. Itraconazole is preferred over fluconazole due to concerns of lower efficacy and development of resistance [98]. Other triazoles, including voriconazole, posaconazole and isavuconazole, have been studied. Voriconazole has been shown to have a low barrier for resistance, while posaconazole remained active in strains with fluconazole resistance [99]. Furthermore, voriconazole has been associated with increased mortality in the first 42 days when compared to itraconazole [100]. Isavuconazole also displayed a higher barrier to resistance and may be effective as step-down oral therapy for histoplasmosis [101]. Patients with mild-to-moderate infection may be treated with itraconazole monotherapy [102].

Conventional itraconazole (referred to here as itraconazole), is only available in oral formulations (i.e., capsules and solution). It has a suboptimal absorption with variable pharmacokinetics [99]. Absorption of oral capsules can be improved by consumption with acidic drinks (e.g., sodas) and fatty meals. Conversely, the oral solution is better absorbed on an empty stomach and is not affected by acid-suppressive therapies. Given the variable pharmacokinetic, therapeutic drug monitoring is recommended in all patients [100]. The metabolism of itraconazole produces an extensive number of metabolites; of those, hydroxyitraconazole has been found to have antifungal activity which is comparable to the parent drug [99]. When measuring serum concentrations by chromatographic tests (high-performance liquid chromatography [HPLC] or liquid chromatography mass spectrometry [LC-MS]), the sum of itraconazole and hydroxyitraconazole provide the total bioactive drug concentration.

The therapeutic target level of itraconazole/hydroxyitraconazole for the treatment of histoplasmosis is unknown, and the current recommendations of maintaining levels of at least 1.0 mcg/mL are derived from extrapolation of other fungal infections [100,101,102]. In patients with oropharyngeal candidiasis, itraconazole serum concentrations <1.0 mcg/mL were associated with treatment failures [103]. A recent study has shown that subtherapeutic concentrations (<1.0 mcg/mL) of itraconazole and hydroxyitraconazole were associated with increased mortality in patients with blastomycosis [104]. It is unclear if a concentration above 1.0 mcg/mL of the parent drug (itraconazole) or a combination of parent drug + metabolite (Itraconazole/hydroxyitraconazole) is required for improved outcomes in the management of histoplasmosis.

A newer formulation of itraconazole labeled super bioavailable (SUBA)-itraconazole has been developed with less variability under fasted condition [105]. However, its use has been limited due to its increased cost.

It is important to assess drug interactions as triazoles are strong inhibitors of the cytochrome P450 isoenzymes, particularly CYP3A4. Careful adjustment and drug level monitoring of other medications, including calcineurin inhibitors is pivotal, as it could lead to supratherapeutic levels [100] (Table 4).

Immune reconstitution syndrome has been observed in solid organ transplant recipients during therapy of Histoplasmosis. Withdrawal of immunosuppression and recovery of CD4+ T cell and CD8+ lymphocytes leads to a proinflammatory state. The management of immune reconstitution syndrome is predominately supportive, though in cases with severe inflammation, corticosteroids can be considered [106,107].

The 2007 Infectious Disease Society of America Histoplasmosis Treatment Guidelines and the American Society of Transplantation Infectious Diseases Community of Practice recommend that therapy should be continued for a minimum of 12 months [97,101]. Assi et al. showed that relapses occur in up to 6% of patients, with two thirds occurring in the first 2 years following diagnosis. Of the patients who relapsed, 50% were treated with less than 7 months of therapy. The only risk factor associated with relapse was failure to reduce calcineurin inhibitor dosage [13].

The 2007 Infectious Disease Society of America Histoplasmosis Treatment Guidelines and the American Society of Transplantation Infectious Diseases Community of Practice recommend monitoring urine and serum *Histoplasma* antigen levels measured by EIA at the time treatment is initiated, at 2 weeks, 1 month, then every 3 months during therapy and up to 6 months following completion of therapy [97,101]. Persistent low-level antigenuria is not infrequent, though patients with a urine *Histoplasma* antigen EIA level of >2 ng/mL at the time of stopping therapy are more likely to relapse [13]. The current guidelines recommend continuation of therapy until a urine *Histoplasma* antigen EIA level of <2 ng/mL is achieved [97,101].

## 7. Novel Antifungal Therapies and Strategies

Recently, the World Health Organization has endorsed a single high-dose regimen of amphotericin B for the treatment of cryptococcal meningitis [108,109]. This regimen is not only associated with fewer side effects, but it also has the advantage of limiting hospitalization costs. A recent prospective randomized multicenter open-label trial comparing a one- or two-dose induction therapy with liposomal amphotericin B versus the standard of care (daily liposomal amphotericin B for 2 weeks) has been reported [110]. The authors used a single dose (10 mg/Kg of LAmB) or two doses (10 mg/Kg of LAmB on D1 and 5 mg/Kg of LAmB on day 3) followed by step-down therapy with itraconazole. The authors concluded that one-day induction therapy with 10 mg/Kg of LAmB was safe and was not inferior to the standard of care. However, this regimen has not been tested in solid organ transplant recipients, and further data from the phase 3 study are warranted before this regimen could be extrapolated to other populations (clinical trial number 05814432).

A new oral formulation of amphotericin B (oral lipid nanocrystal amphotericin B) has been developed. In a recent randomized control trial, Boulware et al. evaluated the antifungal efficacy of oral lipid nanocrystal amphotericin B with flucytosine vs. the standard of care for the treatment of cryptococcal meningitis. They found that the newer oral formulation demonstrated similar antifungal activity, similar survival and was associated with fewer adverse side effects compared to the standard of care [111]. There are no trials assessing its activity for treatment of histoplasmosis.

## 8. Peri-Transplant Donor and Recipient Considerations

Donor-derived histoplasmosis is estimated to occur in approximately 1:10,000 transplants, though this incidence varies according to the endemicity of the area. In a large study of 449 patients in a hyperendemic area that underwent solid organ transplantation, 24% of the recipients were seropositive (M precipitins band or CF titer ≥ 1:8), though there were no post-transplant infections noted at the 16 month follow up. The authors concluded that the risk of reactivation is low and that routine prophylaxis with antifungal agents for those with radiographic or serological evidence of a prior infection (over 2 years) is not indicated [6]. The current guidelines state that active histoplasmosis within 2 years from transplantation may warrant prophylaxis, but did not specify the duration of prophylaxis [112].

A recent report summarized the cases of donor-derived endemic mycoses including seven cases of donor-derived histoplasmosis [113]. None of the donors were known or suspected to have histoplasmosis at the time of donation. In donors in whom there are concerns for active infection (e.g., granulomatous disease or history of histoplasmosis), cultures of the allograft, *Histoplasma* antigen and antibody results should be measured before the organs are procured. If a donor has evidence of active disease, is *Histoplasma* antigen positive in urine or serum, or *Histoplasma* antibody positive by EIA, transplantation should be delayed until there is evidence that the donor does not have active histoplasmosis.

If there is evidence of histoplasmosis in the explanted organ (e.g., positive fungal stains or presence of granulomas), graft cultures and serologies should be performed. If the cultures are negative but the serologies show an H and/or M precipitin band, or CF ≥ 1:32, then the recipient should receive prophylaxis with itraconazole for 3–6 months. If there is only evidence of granuloma and the CF is between 1:8 and 1:16, then urine and serum antigen monitoring every 3 months for 1 year is reasonable. If the donor died due to histoplasmosis (recognized after transplantation has occurred, as otherwise the donation is not advised) or if the cultures or antigens are positive, then the recipient should be treated for at least 1 year [112].

## 9. Conclusions

*Histoplasma capsulatum*, is the etiological agent for histoplasmosis. It is a dimorphic fungus that grows as a mold in the environment and as a yeast in human tissues. It has a broad global distribution with shifting epidemiology, likely due to climate change and anthropomorphic activity. Infection in immunocompetent recipients is usually asymptomatic or with mild respiratory symptoms, though severe forms including progressive disseminated disease can occur. Solid organ transplant recipients are at increased risk for developing symptomatic disease with progressive disseminated histoplasmosis. Antigen detection in bodily fluids remains the most sensitive test for diagnosis. Most data on treatment recommendations have been derived from other populations, including advanced HIV. Treatment is indicated in all solid organ transplant recipients for at least 12 months.

More research is required to test the efficacy of the current treatment recommendations in solid organ transplant recipients and to develop less toxic treatment regimens with fewer drug interactions. Furthermore, further research is needed to tailor the length of therapy in immunocompromised hosts.

## Figures and Tables

**Figure 1 jof-10-00124-f001:**
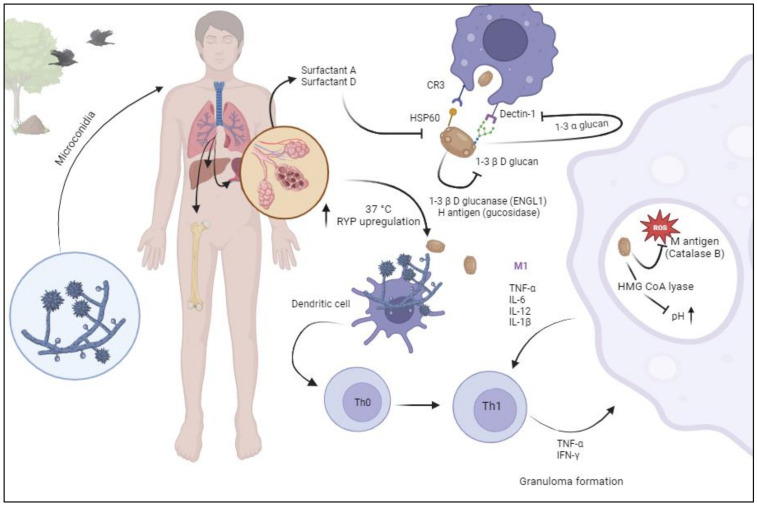
Pathogenesis of histoplasmosis. Aerosolized microconidia are inhaled by the host. Host temperature (37 °C) triggers upregulation of RYP genes, leading to thermal morphological transformation to yeast forms. Once in the alveoli, soluble pathogen recognition receptors (Collectins: surfactant A and D) display a direct fungicidal role through a calcium-dependent mechanism of yeast permeabilization. Heat shock protein 60 (Hsp60) binds to complement receptor 3 (CR3) which prompts phagocytosis. Binding of Dectin-1 and 1-3-β-D-glucan triggers an inflammatory cascade. *Histoplasma* yeast forms, expresses 1-3 α glucan to conceal 1-3-β-D-glucan and expresses 1-3 glucanase and H antigen (glucosidase) to hydrolyzes exposed 1-3-β-D-glucan. Once in the phagolysosome, *Histoplasma* produces catalase B (M antigen) to counteract the radical oxygen species and HMG CoA lyase to present phagolisosome acidification. Dendritic cells interact with naïve T lymphocytes, leading to Th1 polarization. Macrophages can migrate to any part of the reticuloendothelial system and develop granulomas. Image created in BioRender.

**Figure 2 jof-10-00124-f002:**
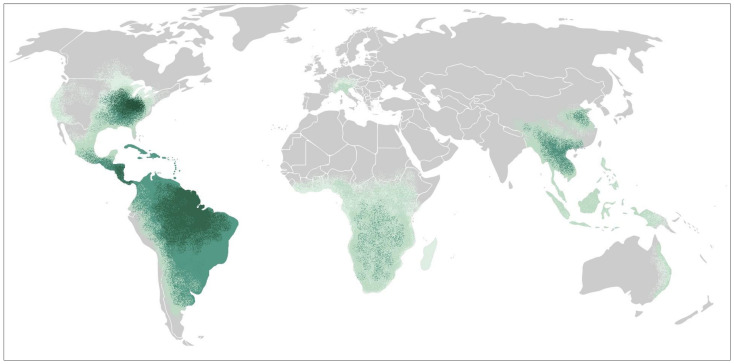
Geographic distribution of histoplasmosis. The dark green represents areas considered hyperendemic, the green represents areas where histoplasmosis infections occur regularly, and the light green represents areas where local infection has been reported.

**Table 1 jof-10-00124-t001:** Clinical manifestations of histoplasmosis in solid organ transplant recipients. Numbers in parenthesis represent the frequency at which the events occur.

Clinical Presentation	Severity of Disease	Clinical Symptoms	Radiographic Changes	Laboratory Abnormalities
Pulmonary (19–36%)	Mild (4–8%)	Fever (90%)	Abnormal chest X ray (50–70%)	Pancytopenia
Acute (symptoms less than 1 month)	Moderate (61–63%)	Shortness of breath (60%)	Abnormal CT chest (87–100%)	Transaminitis
Subacute (symptoms over 1 month)	Severe-Critical (27–35%)	Cough (39%)	Multiple nodules	Elevated lactate dehydrogenase
Chronic (symptoms over 1 month with cavitary lesions)		Diarrhea (35%)	Miliary nodules (6%)	Elevated inflammatory markers
Progressive Disseminated Histoplasmosis (64–81%)		Fatigue	Bibasilar infiltrates	Hypercalcemia
Pulmonary (79–86%)		Malaise	Adenopathy (25%)	Elevated creatinine
Bone marrow (21–37%)		Diaphoresis	Abnormal CT abdomen	
Liver (18–22%)		Headache	Hepatosplenomegaly (25–60%)	
Spleen (9–21%)		Weight loss		
Gastrointestinal (7–12%)		Fever of unknown origin		
Central nervous system (6–9%)				
Skin (2–4%)				

**Table 2 jof-10-00124-t002:** Sensitivity of diagnostic tests for histoplasmosis in solid organ transplant. Numbers with * represent a combination of detection of both MVD EIA antigen and antibody. Numbers with ** represent unpublished data.

	Overall	Pulmonary	Disseminated	CNS
Culture				
Blood	27–63%	49%	90%	
Lung, respiratory	57–72%	60–82%	77%	
Pathology, Cytology				
Lung, respiratory pathology or cytology	77%	86%	74%	
Bone Marrow	71%	-	71%	
Antigen				
MVD Ag EIA				
Urine	85–100%	73%	90–100%	
Blood	86%	50–59%	89–93%	
BAL		83%		
CSF	-	-		78–98% *
IMMY Clarus (urine)		58%	72–91%	
MVD LFA	93%	50%	90–96%	
Serology				
ID and/or CF	33–36%	62–90%	28–71%	51%
MVD IgG/IgM EIA	52% **			82–98% *

**Table 3 jof-10-00124-t003:** Clinical forms and treatment recommendations.

Clinical Form	Treatment Recommendation
**Pulmonary**	
Mild to moderate	Itraconazole for 12 months
Moderately severe or severe	Lipid Amphotericin B for 1–2 weeks followed by Itraconazole for at least 12 months
	and negative or low (<2 ng/mL) urine antigen
	Methylprednisolone 0.5–1 mg/Kg during the first 1–2 if the patient develops ARDS
**Progressive disseminated**	
Mild to moderate	Itraconazole for 12 months
Moderately severe or severe	Lipid Amphotericin B for 1–2 weeks followed by Itraconazole for at least 12 months
	and negative or low (<2 ng/mL) urine antigen
	Methylprednisolone 0.5–1 mg/Kg during the first 1–2 if the patient develops ARDS
**Central nervous system**	
	Lipid Amphotericin B for 4–5 weeks followed by Itraconazole for at least 12 months
	and negative or low (<2 ng/mL) antigen
**Donor derived considerations**	
Active infection	
Died of histoplasmosis	Itraconazole for 12 months
Positive culture and/or positive antigen	
Presence of granuloma or positive serology	Itraconazole for 3–6 months
Negative culture and/or negative antigen	
Pulmonary nodule	No treatment required
without signs of active infection	

**Table 4 jof-10-00124-t004:** Commercially available agents for the treatment of histoplasmosis.

	Usual Dosage	Common Adverse Reactions	Indications for TDM and Timing	Target Trough and Toxicity Concentrations	Suggested Dose Adjustments		Major Drug Interactions	Comments
Triazole								
Itraconazole (oral therapy of choice)								
Conventional capsule and oral solution	Loading dose:	Congestive heart failure	All patients (treatment or prophylaxis)	Level goal: ≥1.0 µg/mL	<0.25 µg/mL	Increase by 50%	ITZ is an inhibitor of CYP3A4	ITZ capsules: Take with fatty food and acidic beverages (e.g., soda)
	200 mg TID × 3 days	GI effects		(Sum of ITZ + Hydroxy-ITZ)	≥0.25–1 µg/mL	Increase by 25%	ITZ will increase the concentration of the following:	Avoid any gastric acid reduction therapy
	Maintenance dose:	Hepatotoxicity	~7 days following loading dose, ~14 days without loading dose				Cyclosporine: Dose of CSA may require 50% reduction	ITZ solution: Take on empty stomach
	200 mg BID	Peripheral edema					Tacrolimus: Dose of TAC may require a one-third reduction	SUBA-ITZ: Take with food
		QT prolongation		Toxicity concentration: Not established			Sirolimus: Dose of SIR may require 50% to 90% reduction	
SUBA-itraconazole	Initial dose:		Sample obtained at any point in the dosing interval (random level)				Rifamycins: Coadministration is usually contraindicated	
	130 mg D (max dose: 130 mg BID)	Headache					Statins: Dose reduction may be required. Monitor for	
		Hearing loss					rhabdomyolysis	
							Calcium channel blockers: Monitor for toxicity	
Fluconazole								
IV or Oral	Induction:	Hepatotoxicity	Not recommended				FCZ is an inhibitor of CYP3A4 and CYP2C9	Single point mutation in CYP51p leads to increased MIC and resistance
	FCZ 800 mg D × 12 weeks	GI effects					CSA, TAC and SIR: Close monitoring of recommended	Exposure to fluconazole has been associated with a reduction in susceptibility to voriconazole in 40% of the isolates
	Maintenance:	QT prolongation					Rifamycins: Coadministration is not advised	
	FCZ 400 mg D	Headaches						Increased failure, relapse rate and mortality compared to ITZ
Voriconazole								
IV	Loading dose: 6 mg/Kg IV BID × 2 doses	QT prolongation	All patients (treatment or prophylaxis)	Trough goal:	<0.5 µg/mL	Increase by 50%	VCZ is an inhibitor of CYP3A4, CYP2C19, CYP2C9	VCZ should be taken on an empty stomach
		Visual disturbances		≥1.0–<5.5 µg/mL	≥0.5–1 µg/mL	Increase by 25%	VCZ will increase the concentration of the following:	Other triazole recommended, particularly in patients with prior fluconazole exposure due to increased risk of resistance
	Maintenance dose: 4 mg/Kg IV BID	Fluorosis	~7 days following loading dose				Cyclosporine: Dose of CSA may require 50% reduction	
		Hepatotoxicity		Toxicity concentrations:	≥1.0–<5.5 µg/mL		Tacrolimus: Dose of TAC may require a one-third reduction	A study showed increased early mortality when compared to ITZ
Oral	Loading dose: 400 mg BID × 2 doses	Skin cancer (long term use)	Sample obtained prior to any dosing (trough level)	≥5.5 µg/mL			Sirolimus: Coadministration is usually contraindicated	Cyclodextrin in IV formulation has not been associated with increased side effects in patients with CrCL < 50 mL/min
		Hallucinations			≥5.5 µg/mL	Decrease by 25%. If severe side effects, hold	Rifamycins: Coadministration is usually contraindicated	
	Maintenance dose: 200 mg BID	GI effects					Statins: Dose reduction may be required. Monitor for	
							rhabdomyolysis	
						therapy until level is <5.5 µg/mL	Calcium channel blockers: Monitor for toxicity	
							Sulfonylurea: Monitor for signs of hypoglycemia	
Posaconazole								
IV	Loading dose:	Hepatotoxicity	All patients (treatment or prophylaxis)	Trough goal:	<1.5 µg/mL	Increase by 33%	PCZ is an inhibitor of CYP3A4	PCZ immediate release suspension:
PO (immediate release suspension or delayed release table)	300 mg BID × 2 doses	GI effects		≥1.5 µg/mL			PCZ will increase the concentration of the following:	Take with fatty food and acidic beverages (e.g. soda)
	Maintenance dose:	QT prolongation	~7 days following initiation of therapy				Cyclosporine: Dose of CSA may require 50% reduction	Avoid any gastric acid reduction therapy
	300 mg D	Headaches		Toxicity concentrations:			Tacrolimus: Dose of TAC may require a one-third reduction	PCZ delayed release tablet:
			Sample obtained prior to any dosing (trough level)	Not clearly stablished			Sirolimus: Dose of SIR may require 50% to 90% reduction	Take with food
				≥3 µg/mL to 3.75 µg/mL			Rifamycins: Coadministration is usually contraindicated	
							Statins: Dose reduction may be required. Monitor for	
							rhabdomyolysis	
							Calcium channel blockers: Monitor for toxicity	
							Sulfonylurea: Monitor for signs of hypoglycemia	
Isavuconazole								
IV or PO	Loading dose:	Shortening of QT interval	Not recommended				ISZ is an inhibitor of CYP3A4	Limited data on efficacy for treatment of histoplasmosis
	372 mg TID × 6 doses	Hepatotoxicity					ISZ will increase the concentration of the following:	
	Maintenance dose:	GI effects					Cyclosporine: Monitor levels	
	372 mg D	Hypokalemia					Tacrolimus: Monitor levels	
		Peripheral edema					Sirolimus: Monitor levels	
							Rifamycins: Coadministration is usually contraindicated	
							Statins: Dose reduction may be required. Monitor for	
							rhabdomyolysis	
							Calcium channel blockers: Monitor for toxicity	
							Sulfonylurea: Monitor for signs of hypoglycemia	
Polyenes								
Amphotericin B (AmB)								
AmB-deoxycholate	0.7–1 mg/kg/day	Acute infusion reactions	Not recommended				Digitalis glycosides	Close monitoring of electrolyte disturbances with aggressive repletion needed
Liposomal AmB	3–5 mg/kg/day	Fever, back pain, hypotension, GI effects					Antiarrhythmic medications	
Lipid-based AmB	5 mg/kg/day							Requires pre- and post- infusion hydration to reduce the risk of nephrotoxicity
		Electrolyte abnormalities						
		Hypokalemia, hypomagnesemia						Infusion-related reactions can be pretreated with antihistamines and antipyretics
		Nephrotoxicity						Infusion-related reactions tend to subside during subsequent infusions

## Data Availability

Data are contained within the article.

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
