# Peer review of "Histoplasmosis in Solid Organ Transplantation"

_jof, 2024, doi:10.3390/jof10020124_

Round 1

Reviewer 1 Report

Comments and Suggestions for Authors

In the manuscript by Barros and Wheat, the authors review the epidemiology, clinical presentation, diagnosis and management of histoplasmosis in solid organ transplant (SOT) patients. The manuscript provides a comprehensive review of histoplasmosis in SOT; however, many sections provide useful information on histoplasmosis in general. The main issue with the manuscript is the frequent citation of other review articles. Some specific suggestions to consider are listed below.  In addition, there are couple minor corrections and suggestions, which can improve the manuscript.

Cited review articles should be replaced with primary literature when possible:

-Line 54, ref. 16

-Line 67, ref. 11

-Line 68, ref. 12

-Line 70, ref. 20

-Line 82, ref. 13

-Line 85, ref. 16

Figure/Table corrections:

-Figure 1 and Figure 2 should be swapped per their mention in lines 105 and 114.

-Table 3 is illegible. Please consider providing it in a landscape format.

-Consider resizing Tables 1 and 2.

Other minor corrections:

-Line 43: “is” should be removed between “this” and “environment”

-Line 122: “temperature” is misspelled

-Line 129: “counteract” is misspelled

-Line 189: “the” should be removed between “and” and “with”

-SOT, please define the abbreviation at the first use.

-Please check 1,3-B-D-glucan and CD4+ throughout the text and be consistent in spelling

-Please check Histoplasma and other genus/species names throughout  and italicize (i.e. lines 127, 302, 333, 334, 338)

-Please check all sentences starting with XYZ et al. and fix them in the format of [Last name of the first author] et al. (i.e. lines 117, 153, 165, 178, 366).

-Line 161, the parentheses should be closed at the end of the sentence.

-Line 257, reference 20 is unrelated here and wrongly cited.

-Line 296, What is (10859)?

-Line 319-321: Please re-word the sentence starting with “Low-level..”

-Line 464, “NCT05814432” can be labeled as “clinical trial number”

-Line 481, “it’s” should be “its”

-Line 531, “further” can be changed to “more” or “future”

Author Response

In the manuscript by Barros and Wheat, the authors review the epidemiology, clinical presentation, diagnosis and management of histoplasmosis in solid organ transplant (SOT) patients. The manuscript provides a comprehensive review of histoplasmosis in SOT; however, many sections provide useful information on histoplasmosis in general. The main issue with the manuscript is the frequent citation of other review articles. Some specific suggestions to consider are listed below.  In addition, there are couple minor corrections and suggestions, which can improve the manuscript.

Cited review articles should be replaced with primary literature when possible:

-Line 54, ref. 16

            Switched for Beyhan et al. A temperature-responsive network links cell shape and virulence traits in a primary fungal pathogen.

-Line 67, ref. 11

            Switched for Benedict et al. Epidemiology of Histoplasmosis Outbreaks, United States, 1938-2013.

-Line 68, ref. 12

            Switched for Assi et al.

-Line 70, ref. 20

            This is an excellent review article that describes different aspects of dectin-1. It is more complete than a primary reference.

-Line 82, ref. 13

            Switched for Fisher et al. Determination of beta-glucosidase enzymatic function of the histoplasma capsulatum H antigen using a native expression system.

-Line 85, ref. 16

            We have added 3 references. We have kept that review article as it is excellent.

Figure/Table corrections:

-Figure 1 and Figure 2 should be swapped per their mention in lines 105 and 114.

            We have swapped the figures.

-Table 3 is illegible. Please consider providing it in a landscape format.

            We apologize, it was supposed to be on a landscape format.

-Consider resizing Tables 1 and 2.

            We have increased their sizes for easier visualization.

Other minor corrections:

-Line 43: “is” should be removed between “this” and “environment”

            We have removed “is”

-Line 122: “temperature” is misspelled

            We have fixed this misspelling.

-Line 129: “counteract” is misspelled

            We have fixed this misspelling.

-Line 189: “the” should be removed between “and” and “with”

            We have removed “the”

-SOT, please define the abbreviation at the first use.

            Line 153: we have added solid organ transplant (SOT)

-Please check 1,3-B-D-glucan and CD4+ throughout the text and be consistent in spelling

            All 1-3-B-D-glucan are now spelled consistently throughout the manuscript (we have introduced the same spelling for 1-3-A-glucan.

            All CD4+ T cells are now spelled consistently throughout the manuscript.

-Please check Histoplasma and other genus/species names throughout and italicize (i.e. lines 127, 302, 333, 334, 338)

            We have italicized all mentions of Histoplasma and other genus/species.

-Please check all sentences starting with XYZ et al. and fix them in the format of [Last name of the first author] et al. (i.e. lines 117, 153, 165, 178, 366).

            We have fixed the format in all of the sentences that include XYZ et al.

-Line 161, the parentheses should be closed at the end of the sentence.

            We have added the parenthesis at the end of the sentence.  

-Line 257, reference 20 is unrelated here and wrongly cited.

            We apologize for that mistake. We have fixed that.

-Line 296, What is (10859)?

            We have deleted that number and added a reference.

-Line 319-321: Please re-word the sentence starting with “Low-level..”

            Now states: “Low-level cross-reactivity occurs with the galactomannan detected in coccidioidomycosis and other dimorphic fungi.”

-Line 464, “NCT05814432” can be labeled as “clinical trial number”

            Now states: (clinical trial number 05814432). 

-Line 481, “it’s” should be “its”

            We have fixed this error.

-Line 531, “further” can be changed to “more” or “future”

                We have changed it to “more”

Reviewer 2 Report

Comments and Suggestions for Authors

In this review article on histoplasmosis infection in solid organ transplantation the authors discuss the pathogenesis, epidemiology, clinical manifestations and management of histoplasmosis in solid organ transplant recipients. The manuscript is in general well written and structured. The figures and tables enhance the readability of the manuscript.

The topic is suitable for the Special Issue” Histoplasma and Histoplasmosis” However, in my opinion, some points need to be addressed before publication as follows:

1.       The abstract is too narrow and does not contain enough information.

2.       Line 20: The authors stated that there are three varieties of Histoplasma. However, this classification should no longer be used since H. capsulatum is currently considered a complex of cryptic species, consisting of several groups of isolates that differ genetically and they correlate with a particular geographic distribution. Please see and cite the following article “V.E. Sepúlveda et al.  Genome sequences reveal cryptic speciation in the human pathogen Histoplasma capsulatum. Mbio, 8 (2017), p. e01339-17”.

3.       Methodology section is missing. What criteria did the authors select for their review article? Methodology section is not mandatory for narrative reviews but it may improve the quality of the manuscript.

4.       It is not clear if the tables are taken from another source. If so, please specify it.

5.       In the paragraph 5.6, with regard to the antibody detection, the authors should also discuss that antibodies are detectable only 4-8 weeks after initial infection and this represents the major limitation of the antibody test for histoplasmosis, especially among immuno-compromised individuals who are unable to generate an effective immune response to H. capsulatum antigens.  Please see the article “Azar, M. M.; Hage, C. A. Laboratory diagnostics for histoplasmosis. Journal of Clinical Microbiology 2017, 55(6), 1612–1620 (Ref. 77).

6.       In the paragraph 5.7, regarding the statment “MALDI-TOF MS has been shown to be highly accurate for the identification of H. capsulatum from cultures” (lines 373), the authors should specify that this technique is a useful tool for Histoplasma identification in endemic areas where the diagnosis may be performed by using MALDI-TOF instruments with the Vitek MS v3.0 database. Contrary, in non endemic areas, the use of MALDI-TOF with the Bruker Daltonics software does not allow the identification of dimorphic fungi, including H. capsulatum, due to the lack of the fungal reference spectrum in its database. Please see the article ”Cosio T. et al. Closing the Gap in Proteomic Identification of Histoplasma capsulatum: A Case Report and Review of Literature. Journal of fungi (Basel, Switzerland), 9(10), 1019, 2023.  doi.org/10.3390/jof9101019”.

7.       Please carefully proof-read spell check to eliminate grammatical errors

Minor comments

Line 43 “ This is environment is present” please correct  “This  environment is present”

Line 88  please check the sentence “Macrophages play a dual role by containing an infection and providing an environment where replication and/or dormancy” . It is incomplete.

Line 117 “United states” “United States”

Line 140 “a low-level infection “ Please, specify that the fungus can persist into the granuloma

Line 153 “SOT” please the first time provide the complete term and enclose the abbreviation in parentheses.

Line 153 “Assi et” “Assi et al.”

Line 189 “non-specific and the with a wide range..” delete “the” before with

Line 195 “Number in parenthesis” “Numbers in paranthesis”

Line 294 “galactomannan and tested in body fluids” please delet “and “ before tested

Line 336 “serum histoplasma” “serum Hstoplasma”

Line 480 please check the sentence “The current guidelines that active histoplasmosis within 2 years from trans plantation may warrant prophylaxis…” The verb is missing

Figure 1 legend: The dark green represent  “represents”; “the green represent” “represents”; “light green represent” “represents”

Figure 2 legend “Histoplasma” in italic; line 131 “ leading Th1 polarization” “leading to Th1 polarization”

Comments on the Quality of English Language

Minor  English editing

Author Response

In this review article on histoplasmosis infection in solid organ transplantation the authors discuss the pathogenesis, epidemiology, clinical manifestations and management of histoplasmosis in solid organ transplant recipients. The manuscript is in general well written and structured. The figures and tables enhance the readability of the manuscript.

The topic is suitable for the Special Issue” Histoplasma and Histoplasmosis” However, in my opinion, some points need to be addressed before publication as follows:

  1. The abstract is too narrow and does not contain enough information.

We have expanded the abstract.

  1. Line 20: The authors stated that there are three varieties of Histoplasma. However, this classification should no longer be used since H. capsulatum is currently considered a complex of cryptic species, consisting of several groups of isolates that differ genetically and they correlate with a particular geographic distribution. Please see and cite the following article “V.E. Sepúlveda et al.  Genome sequences reveal cryptic speciation in the human pathogen Histoplasma capsulatum. Mbio, 8 (2017), p. e01339-17”.

      We have included this and mention that we are using the historical classification. We acknowledge that it is incomplete and old though the clinical data was obtained prior to the newer classification.

  1. Methodology section is missing. What criteria did the authors select for their review article? Methodology section is not mandatory for narrative reviews but it may improve the quality of the manuscript.
  2. It is not clear if the tables are taken from another source. If so, please specify it.

      We have created the tables. They are not taken from anywhere else

  1. In the paragraph 5.6, with regard to the antibody detection, the authors should also discuss that antibodies are detectable only 4-8 weeks after initial infection and this represents the major limitation of the antibody test for histoplasmosis, especially among immuno-compromised individuals who are unable to generate an effective immune response to H. capsulatum antigens.  Please see the article “Azar, M. M.; Hage, C. A. Laboratory diagnostics for histoplasmosis. Journal of Clinical Microbiology 2017, 55(6), 1612–1620 (Ref. 77).

We have included a line mentioning that limitation

  1. In the paragraph 5.7, regarding the statment “MALDI-TOF MS has been shown to be highly accurate for the identification of H. capsulatum from cultures” (lines 373), the authors should specify that this technique is a useful tool for Histoplasma identification in endemic areas where the diagnosis may be performed by using MALDI-TOF instruments with the Vitek MS v3.0 database. Contrary, in non endemic areas, the use of MALDI-TOF with the Bruker Daltonics software does not allow the identification of dimorphic fungi, including H. capsulatum, due to the lack of the fungal reference spectrum in its database. Please see the article ”Cosio T. et al. Closing the Gap in Proteomic Identification of Histoplasma capsulatum: A Case Report and Review of Literature. Journal of fungi (Basel, Switzerland), 9(10), 1019, 2023.  doi.org/10.3390/jof9101019”.

We have included this in the paragraph.

  1. Please carefully proof-read spell check to eliminate grammatical errors

Minor comments

Line 43 “ This is environment is present” please correct  “This  environment is present”

We have removed “is”

Line 88  please check the sentence “Macrophages play a dual role by containing an infection and providing an environment where replication and/or dormancy” . It is incomplete.

            Now states: “Macrophages play a dual role by containing an infection and providing an environ-ment where replication and/or dormancy can occur. H. capsulatum uses them as a trojan horse for dissemination.”

Line 117 “United states” “United States”

            We have fixed this issue.

Line 140 “a low-level infection “ Please, specify that the fungus can persist into the granuloma

Line 153 “SOT” please the first time provide the complete term and enclose the abbreviation in parentheses.

            Line 153: we have added solid organ transplant (SOT)

Line 153 “Assi et” “Assi et al.”

            We have fixed all the XYZ et al. in the manuscript.

Line 189 “non-specific and the with a wide range..” delete “the” before with

            We have removed “the”

Line 195 “Number in parenthesis” “Numbers in parenthesis”

            We have changed it to plural “Numbers in parenthesis”

Line 294 “galactomannan and tested in body fluids” please delet “and “ before tested

            We have deleted “and”

Line 336 “serum histoplasma” “serum Hstoplasma”

            We have fixed this error. In addition, all Histoplasma are now italicized.

Line 480 please check the sentence “The current guidelines that active histoplasmosis within 2 years from trans plantation may warrant prophylaxis…” The verb is missing

            Now states: “The current guidelines state that active histoplasmosis within 2 years from transplanta-tion may warrant prophylaxis, but did not specify the duration of prophylaxis”

Figure 1 legend: The dark green represent  “represents”; “the green represent” “represents”; “light green represent” “represents”

            We have fixed this.

Figure 2 legend “Histoplasma” in italic; line 131 “ leading Th1 polarization” “leading to Th1 polarization”

            We have italicized Histoplasma and also added “to” to Th1 polarization.

Reviewer 3 Report

Comments and Suggestions for Authors

This is a useful, broadly based, expert review of histoplasmosis, centered on disseminated H. capsular var. capsulatum infection and  details of the authors  company's diagnostic tests. There is an appropriate discussion about solid organ transplantation recipients and donor derived infection. I was interested to note that their discussion didn't include immune reconstitution syndrome or HLH. Though it certainly wasn't essential, those topics would have fit into their detailed discussion of the immune response.

Author Response

This is a useful, broadly based, expert review of histoplasmosis, centered on disseminated H. capsular var. capsulatum infection and details of the authors company’s diagnostic tests. There is an appropriate discussion about solid organ transplantation recipients and donor derived infection. I was interested to note that their discussion didn't include immune reconstitution syndrome or HLH. Though it certainly wasn't essential, those topics would have fit into their detailed discussion of the immune response.

 We appreciate the comments. We have included the discussion of IRS and HLH.

Round 2

Reviewer 2 Report

Comments and Suggestions for Authors

In this revised version the quality of manuscript has been improved and therefore I consider it acceptable for publication.